# Convolutional Neural Network to Classify Infrared Thermal Images of Fractured Wrists in Pediatrics

**DOI:** 10.3390/healthcare12100994

**Published:** 2024-05-11

**Authors:** Olamilekan Shobayo, Reza Saatchi, Shammi Ramlakhan

**Affiliations:** 1Department of Computing, Sheffield Hallam University, Sheffield S1 2NU, UK; olamilekan.s.shobayo@student.shu.ac.uk; 2Department of Engineering and Mathematics, Sheffield Hallam University, Sheffield S1 1WB, UK; 3Emergency Department, Sheffield Children’s Hospital NHS Foundation Trust, Sheffield S10 2TH, UK; sramlakhan@nhs.net

**Keywords:** convolutional neural network, deep learning, medical infrared imaging and classification, bone fracture identification

## Abstract

Convolutional neural network (CNN) models were devised and evaluated to classify infrared thermal (IRT) images of pediatric wrist fractures. The images were recorded from 19 participants with a wrist fracture and 21 without a fracture (sprain). The injury diagnosis was by X-ray radiography. For each participant, 299 IRT images of their wrists were recorded. These generated 11,960 images (40 participants × 299 images). For each image, the wrist region of interest (ROI) was selected and fast Fourier transformed (FFT) to obtain a magnitude frequency spectrum. The spectrum was resized to 100 × 100 pixels from its center as this region represented the main frequency components. Image augmentations of rotation, translation and shearing were applied to the 11,960 magnitude frequency spectra to assist with the CNN generalization during training. The CNN had 34 layers associated with convolution, batch normalization, rectified linear unit, maximum pooling and SoftMax and classification. The ratio of images for the training and test was 70:30, respectively. The effects of augmentation and dropout on CNN performance were explored. Wrist fracture identification sensitivity and accuracy of 88% and 76%, respectively, were achieved. The CNN model was able to identify wrist fractures; however, a larger sample size would improve accuracy.

## 1. Introduction

Children may sustain wrist fractures from a range of injury mechanisms. They are more at risk of harm from X-ray radiation than adults [1]. Around 50% of X-ray radiographs obtained to diagnose wrist fractures do not show a fracture. In addition, serial X-rays are often obtained to identify healing fractures where they are not visible initially. Therefore, a harmless, easy-to-use and cost-effective means of identifying wrist fractures at the index presentation could reduce the number of unnecessary X-rays.

Interests in the medical diagnostic and monitoring applications of infrared (IR) thermal (IRT) imaging have grown in recent years, as infrared sensors and image processing techniques have improved significantly. A modern IRT camera produces images by accurately measuring the heat emanating from objects in the form of IR [2]. A feature of IRT imaging is its noncontact nature, allowing measurements without disturbance to the individual or risk of contamination [3]. IRT imaging has proved effective in biological applications [3] and the identification of pediatric bone fractures [1,4]. An analysis of IRT images demonstrated a statistically significant temperature difference between the fractured wrists and their contralateral (uninjured) wrists caused by the changes in the blood flow at the injury site [1]. The purpose of this study was to develop a deep learning (DL) CNN to differentiate between wrists with a fracture and wrists without a fracture, i.e., wrists with a sprain.

### 1.1. Deep Learning

Deep Learning (DL) can identify the distinguishing features of images for their differentiation by using multiple layers of non-linear transformations called hidden layers. DL performs its feature extraction operation in an automated manner compared with pre-processing methods such as principal component analysis that perform dimensionality reduction [5]. It learns by simulating the neurons in the human brain to process substantial volumes of data [6]. There have been extensive developments in deep learning in recent years with applications such as computer vision, natural language processing and speech recognition [7]. DL has emerged from the conventional artificial neural network concept, but it can considerably outperform earlier models by accurately analyzing large complex information. DL approaches can be classified by the manner in which they learn as being supervised, unsupervised or partially supervised [8]. They can also be classified by the type of network as being recursive neural networks, recurrent neural networks, and convolutional neural networks (CNNs) [8]. DL algorithms have been used increasingly for classification tasks that include image recognition. This is due to the non-linear complexities that some image classifications can pose and the multilayered adaptivity of DL to handle these types of classifications. The initial layers are unsupervised in nature, allowing the DL architecture to learn the image features, with the final layers being more supervised to help with the classification process [9].

Feature extraction has traditionally been dependent on prior knowledge of the modeler with the use of statistical techniques [10]. However, these require judgment of the suitability and effectiveness of differentiating features [9].

### 1.2. Convolutional Neural Network and Data Augmentation

The convolutional neural network (CNN, Figure 1) is a type of DL for image processing that avoids manual feature extraction [11].

Using the ImageNet datasets, which are widely used in industry and academia, CNNs have achieved a remarkable capability for both image classification and pattern recognition tasks. The working principle is usually the same and is composed of convolution layers applied at the input of a supervised DL network that updates its weights of feature maps, learnt from the previous layer, based on their relationships in space to reduce the numbers of parameters needed to train the model [6,12,13]. This is then followed by fully connected layers and the output layer, which could be either classification (SoftMax layer), recognition or clustering.

There are several variants of CNN models commonly referred to as transfer learning (TL) models. These include ImageNet, LeNet, Inception, VGGNet and ResNet [10]. These models have been trained on a very large number of high-resolution images from the internet known as the ImageNet with millions of trainable parameters, with their weights and biases frozen. The input and classification layers can be adjusted to train other types of healthcare-related images such as those from computerized tomography scans, IR thermography and magnetic resonance imaging.

A challenge encountered in applying DL techniques to medical applications is small datasets, a problem known as the data challenge [14]. A method to deal with it is called data augmentation [15]. Data augmentation could allow DL models to improve their data classification performance by artificially generating a greater diversity of training examples [15,16,17]. Therefore, augmentation techniques are transformation operations that modify a sample (e.g., an image) in such a way that it can still be semantically described by the original identity of the sample category or type [18]. Data augmentation has been applied as part of Raman spectroscopy skin cancer tissue classification [19], natural language processing [20], time series classification with neural networks [21], text classification [22], material microscopic image segmentation [23] and image classification [24]. The were also studies outlining the effectiveness of generative adversarial networks [25,26] in medical image augmentation [27,28,29,30].

A further challenge when using medical data for machine learning diagnostic purposes may be a significant imbalance in the number of samples (example cases) from different categories. This imbalance may bias the analysis and thus affect the accuracy of the results. A study reported a means to deal with this imbalance by using Euclidean distance between samples of the same category to reduce the associated imbalance [31].

The contributions of this study include the following:A CNN model devised to identify wrist fractures from IRT images. The ability of the CNN was leveraged to automatically extract features based on the convolution layers as previous methods used manual feature extraction methods.A proposed CNN model that was tailor-made for the specific input IRT image characteristics.An exploration of the effectiveness of image augmentation when developing DL models for IRT image classification was performed.The performance of CNN in identifying wrist fractures was compared with an earlier study that used a multilayer perceptron neural network, indicating improved accuracy for the CNN.

## 2. Related Studies

In this section, studies utilizing infrared thermal imaging for medical diagnosis are initially described and then applications of machine learning to classify infrared thermal images are provided.

### 2.1. Infrared Thermal Imaging for Medical Diagnosis

Several studies have used IRT imaging to diagnose or monitor medical conditions. In a diagnostic pilot study [1], IRT imaging was used to differentiate between wrist fractures and sprains (no fracture) in children. The temperature of the injured wrist, represented by the relevant region of interest (ROI), was compared with the temperature of the contralateral uninjured wrist acting as its control. Statistical techniques were used to analyze the temperature difference. The study indicated that there was a significant temperature difference, with higher values recorded in the fractured wrists than in the sprained wrists. Infrared thermal imaging has been studied to identify fractured thoracic vertebrae in children aged 5–18 years (number: 11) with osteogenesis imperfecta (OI) [4]. OI is a genetic disorder causing bones to become more fragile and thus more prone to fractures. IRT imaging provided a cost-effective and quick (as compared with magnetic resonance imaging or computerized tomography) method of detecting thoracic vertebrae. A summary of other related studies using IRT imaging for medical diagnosis is provided in Table 1.

### 2.2. Machine Learning and Deep Learning Techniques

Machine learning classification techniques can provide means of computer-aided medical diagnosis [36,37]. A multilayer perceptron (MLP) artificial neural network (ANN) model for wrist fracture classification using IRT images of 40 subjects was developed [38]. The features used for training the MLP were extracted manually, which comprised the pixel values of the IRT images extracted from the wrist ROI. A discrimination accuracy of 77% was obtained. In another study, a custom deep learning neural network (DNN) algorithm was developed to classify obesity using thermal images of the abdomen, forearm, and shank region as the ROI [36]. They recruited 50 obese and 50 healthy patients. Their CNN-based method provided an accuracy of 92%. In another study, different artificial intelligence (AI) techniques were compared to classify IRT images of 39 patients with diabetic foot ulcers [39]. The images were processed for classification by three decision support machine learning algorithms, namely ANN, support vector machine (SVM) and k-nearest neighbors (k-NNs). Their comparisons of these techniques indicated that the k-NNs of five neighbors provided the best classification of the disease with an accuracy of 81.25%. Multiple CNN models were examined to differentiate cancerous blood cells from normal blood cells and the best results were obtained through a majority voting scheme [40]. Studies employed DL as part of detecting COVID-19 [41,42]. CNN models were devised to classify IRT images of fatty liver [43]. They used three types of datasets based on specified exclusion criteria. Their best approach provided a classification accuracy of 94%. A CNN-based transfer learning algorithm called ResNet50 was used to classify thermal images of pneumonia from COVID-19 from 101 patients, providing an accuracy of 91% [44]. Their training and test images were randomly selected from a pool of data which consisted of several images from the same patient. A cross-validation technique was used to evaluate the model.

Other related studies that used computer-aided screening for various medical conditions included predicting hemodynamic shock [45], diagnosis of cardiovascular disease [46], diagnosis of diabetes [47] and radiological applications [48]. A summary of further related studies is provided in Table 2.

## 3. Materials and Methods

The method used in this study comprised data collection and data pre-processing which included fast Fourier transformation (FFT) of the IRT images, image resizing and augmentation. These were followed by classification based on a CNN-based DNN. The operations are outlined in Figure 2.

### 3.1. Participants’ Recruitment and Data Recording

IRT images from 40 participants were used for this study. They comprised 21 confirmed patients without a fracture and 19 patients with a fracture, confirmed by X-ray radiography. For both injured and uninjured wrists, 299 images were recorded (frame rate = 30 frames per second). Further details of the recruitment and recording can be found in [1,38]. The study was conducted in accordance with the Declaration of Helsinki, and approved by the National Health Service Research Ethics Committee (United Kingdom, identification number: 253,940, approval date: 7 March 2019). Informed consent was obtained from all participants included in the study.

### 3.2. Image Pre-Processing

As there can be variations in skin temperature across participants, the mean temperature of the uninjured wrist region of interest (ROI) was subtracted from each pixel value of the injured wrist ROI. The resulting ROIs for the injured wrists were fast Fourier transformed (FFT). Since the wrist ROI varied in dimension across the participants, the FFT process provided a means to resize the images by selecting a section from its center with a size of 100 × 100 pixels and thus ensuring they all had identical dimensions. This section had the main frequency components. An IR image participant’s hands with a fracture of the left wrist is shown in Figure 3a. The fractured left wrist appears brighter in comparison with the uninjured right wrist, indicating an increase in IR thermal emission. An IR image of a participant’s hand with a sprained left wrist is provided in Figure 3b. The temperature differences between the injured and uninjured wrist require image processing analyses to interpret their significance.

The ROIs for the injured left wrists are shown by the blue dotted lines on the respective images. A similar ROI was selected from the uninjured wrist. Typical magnitude frequency spectra of the ROI for fractured and sprained wrists are shown in Figure 4a and Figure 4b, respectively.

The transformed and resized images were then converted into the portable network graphic (PNG) format for CNN input compatibility in preparation for the next stage.

#### Image Augmentation

Geometric transformation methods of rotation, translation, and shearing were performed on the fast-Fourier-transformed and resized ROI sections (images) of the injured wrists. This was to introduce further variabilities in the images to enhance CNN generalization during training [53]. Each image was randomly subjected to a single transformation scheme. For those images that were subjected to rotation transformation, the amount of the rotation was randomly selected to be between −90° and 90°. A lower value of rotation was not as effective for classification. For the images that were translated, the amount of translation was randomly selected to be between −3 pixels and +3 pixels along the horizontal and vertical axes. The translation allowed the input images to have variations in space and avoid positional bias during training. Finally, for the images that were subjected to shearing transformation, the amount of shearing was randomly selected to be between a shear factor of −2 and 2, both horizontally and vertically. Shearing affected the shape of the input images by slanting or tilting them in the specified direction. The values chosen for the translation and shearing were kept quite small as the input data had minimal variations from the point of recording.

### 3.3. CNN-Based Deep Learning

The computer used to perform the processing operations was an Apple MacBook, with an 8-core CPU, 10-core GPU, 16-core neural engine, 100 GB/s memory bandwidth, and maximum CPU clock rate of 3.49 GHz. The CNN training took 104 min and 41 s.

The CNN-based deep neural network was based on the architecture described in Section 1.2. The structure of the CNN-based deep learning neural network architecture is shown in Figure 5. The layers with learnable parameters used in the CNN-based DNN architecture for the classification of the images representing fracture and non-fracture are provided in Table 3. The hyperparameters used for fine tuning the model are provided in Table 4.

The CNN-based deep neural network consisted of 34 layers in total and 3.9 million learnable parameters, i.e., weights and biases. It comprised a similar combination of convolutional, batch normalization, rectified linear unit (ReLU) or rectifier activation function (this introduced non-linearity to the deep learning model) and maximum pooling layers, respectively (they helped in the extraction of relevant features from the input images while simultaneously reducing the data dimension).

The CNN-based deep neural network had 8 convolutional layers. The first layer of convolution comprised 8 filters, with a filter size of 3, connected to the input layer. The ‘convolution2dlayer’ from the Matlab^©^ DL toolbox (Version 2023b) [54] was used to create the CNN model. It defined the filter, referred to as the kernel or mask. It used an array of defined weights applied to the neighboring pixels of the input image while the convolution process was being performed. During the convolution, the filters were arranged at each position of the image input around the pixels centered in the position. There was an element-wise multiplication of the neighboring pixels of the input image and the results of the multiplication were then summed to generate the filter output. This was then slid through the entirety of the image with the convolution process at each position of the window. The output image, also known as the feature map, had the same dimension as the input image. Its values were determined by the convolution operation.

As the layers deepened, the number of convolutional filters increased by 2^(*n*+2)^, where *n* is the CNN convolution layer number, until it reached 512 filters, which was the maximum number of filters that could be achieved based on the size of the input image, as the choice of the convolutional filters must be less than the input image size. As the layer number increased, it enhanced the ability of the CNN model to distinguish features of the images [55].

A batch normalization layer followed each convolutional layer. The batch normalization layer sped up the CNN convergence by normalizing the values of the calculated weights and biases from the previous convolution layer to have a mean value close to zero and a standard deviation close to 1 before passing to the next layer of the CNN.

A rectified linear unit (ReLU) activation layer followed each batch normalization layer. The choice of the ReLU activation function was based on the non-linear classification task that was required of the input images. Each ReLU layer ensured that neurons with negative values remained inactive, allowing neurons with positive values to be activated for the next convolution layer. This helped the CNN model to learn only the main image features, thereby reducing overfitting by the model during its training [56].

The final convolution layer was connected to a fully connected (FC) layer of processing neurons with two outputs representing the injury types, i.e., fracture and non-fracture. The output vectors from the FC layer comprised positive and negative values.

The next layer was the SoftMax layer that converted the output from the FC layer into normalized class probabilities [57]. The classification layer generated the classification of each image based on the probability of injury being a fracture or a non-fracture (sprain) from the SoftMax layer.

The CNN model was trained on 250 epochs (each epoch was 49 iterations) without early stopping, where the final model metrics were calculated based on the average validation metrics throughout the specified number of epochs. The training process was examined for early stopping at 60 epochs (corresponding to 2940 iterations) where there was a global minimum in terms of the training error. As, at that point, the differentiation accuracy was not sufficiently high, the training was allowed to continue for 250 epochs. Thereafter, there was no further improvement in the CNN classification accuracy.

Figure 6 and Figure 7, respectively, show the training and loss (i.e., error) performances of the CNN model with the validation dataset. The blue graph in Figure 6 represents the training of the model, with the black graph representing the model’s validations using the validation data with validation performed after 250 epochs. The red graph in Figure 6 shows the training loss and the black graph shows the validation error.

### 3.4. Evaluation Metrics

The efficacy of the proposed CNN-based DL neural network was assessed using the confusion matrix, which provided classification accuracy, sensitivity, specificity, positive predictive value (PPV) and negative predictive value (NPV). These metrics were obtained from the model’s classification of true negatives (TNs), true positives (TPs), false positives (FPs) and false negatives (FNs). The receiver operating characteristic (ROC) was also used to evaluate the model, providing the area under curve (AUC) for clinical use.

## 4. Results

Two variants of the CNN-based DNN architectures were investigated. The first variant performed classification with image augmentation without a dropout layer, and the second variant did not include augmentation but had a dropout layer.

When training a DL model on small datasets, there is a tendency for overfitting to occur due to the large number of layers used in the network. To deal with this effect, techniques such as augmentation [58] and regularizations (dropout) [59] have been reported. In the first experiment, the images were randomly rotated, translated, and sheared as described in the Materials and Methods section. For the second experiment, a dropout layer was added to the to the last convolution layer, just after the maximum (max) pooling layer and before the fully connected (FC) layer. A value of 0.2 was selected for the dropout layer, meaning that 20% of the nodes were dropped with every update in weight during training. A larger value of dropout would have negatively affected the performance of the model as more nodes would have been dropped.

Binary classifications were provided for fractures and non-fractures (sprains) and presented for both experiments. The dataset comprised 11,960 (i.e., 40 participants × 299 images) fast-Fourier-transformed IRT images of 40 participants, 21 without a fracture (sprain) and 19 with a fracture. The dataset was split into 70% of the images for training (13 participants with a fracture and 15 participants without a fracture, totaling 8372 images) and 30% of the images for validation (6 fractures and 6 non-fractured, totaling 3588 images). The number of images for training and validation is summarized in Table 5.

### 4.1. CNN-Based Deep Neural Network with Augmentation and without Dropout

The CNN-based deep neural network model (Figure 5) was configured based on the hyperparameters provided in Table 4. This model was constructed using the Matlab© DL toolbox [54]. To indicate the effectiveness of the model, the average values of sensitivity, specificity, NPV, PPV and accuracy are provided in Table 6 and the confusion matrix is provided in Figure 8. The receiver operating characteristic curve (ROC) [60] is shown in Figure 9 with the model operating points for fracture and non-fracture. The CNN-based deep neural network model achieved an area under the ROC curve (AUC) of 0.82.

### 4.2. Results for CNN-Based Deep Neural Network without Augmentation and with Dropout

In this part of the study, the same hyperparameters in Table 4 and the architecture in Figure 5 were used, but without image augmentation. To compensate for overfitting, a dropout layer (0.2) was introduced, as discussed earlier. The sensitivity, specificity, NPV, PPV and accuracy are provided in Table 7.

## 5. Discussion

The purpose of this study was to develop an IRT image classification model utilizing a CNN-based deep neural network, in contrast to the wrist fracture identification in [38] where feature extraction for the multilayer perceptron artificial neural network was performed prior to the neural network classification. From the confusion matrix in Figure 8, the model produced an overall accuracy of 76% for the classification of the IRT images between wrist fracture and non-fracture (sprain) when image augmentation was applied without dropout. This provided improved accuracy compared to when a dropout layer was introduced without augmentation. The model also showed an NPV of 80% and a PPV of 73%. Therefore, the classification of images associated with the non-fractured wrists was 80% correct.

The ROC curve presented in Figure 9 indicated the model’s operating point of 0.82 which provided a better measure of classification considering a larger number of non-fractured images in the dataset. The ROC area under the curve (AUC) was used to further highlight the ability of the model in differentiating the injury types. It provided a benchmark for clinical diagnosis regardless of the number of classes. Acceptable values for the AUC ROC are between 0.8 and 1 [60].

When the model was used without augmentation and with a dropout value of 0.2, the model produced a specificity of 68% and a sensitivity of 50%. These values were much lower compared to the first experiment. This can be attributed to the nature of the IRT images used in the CNN architecture.

In a follow-up work, other computer vision techniques such as vision transformers [61] will be considered to improve the classification of IRT images. The effectiveness of other image augmentation techniques and generative models like variational autoencoders [62] and generative adversarial networks [63,64] will also be evaluated to identify wrist fracture IRT images. The original images had small variations, and dropping out played a significant role in the model accuracy as pattern recognition became harder. The inclusion of the image augmentation technique increased the variability of the IRT dataset, assisting CNN generalization during its training.

The main limitation of this study was the inclusion of a small sample size of recruited participants. The augmentation process was adapted to increase the training and validation datasets, as otherwise, given the number of CNN layers, overfitting would have occurred during training. An issue that was carefully considered during both CNN training and validation was the manner in which the augmented images were used as part of the training and validation datasets. There were 299 augmented images from each participant. It was ensured that the augmented images from a single participant were not split between the training and validation datasets. If the augmented images from a single participant were split between the training and validation datasets, the validation accuracy could have unduly increased as the CNN had already “seen” the related images from the same participant during its training phase.

This study carried out a successful development and evaluation of CNN models to interpret augmented IR thermal images for differentiating between wrist fractures and wrist sprains in pediatric patients.

## 6. Conclusions

This study developed a CNN-based deep neural network architecture to differentiate fractured and non-fractured wrists using infrared thermal (IRT) images of 40 participants (11,960 images). The IRT images were fast Fourier transformed to aid their representation and resized in preparation for the CNN model. Augmentation was performed to improve the generalization of the model during training, thereby reducing overfitting by the model. The model performance was examined in two scenarios. With image augmentation and no dropout, the model achieved accuracy, sensitivity, specificity, NPV and PPV values of 76%, 88%,68%, 80% and 73%, respectively. An ROC-AUC of 0.82 was also obtained. For the second scenario, where regularization of the model was performed by including a dropout (0.2) layer for the CNN-based deep neural network architecture, the values of the evaluation metrices decreased. This study was successful in using artificial intelligence-assisted IRT images to identify wrist fractures.

## Figures and Tables

**Figure 1 healthcare-12-00994-f001:**
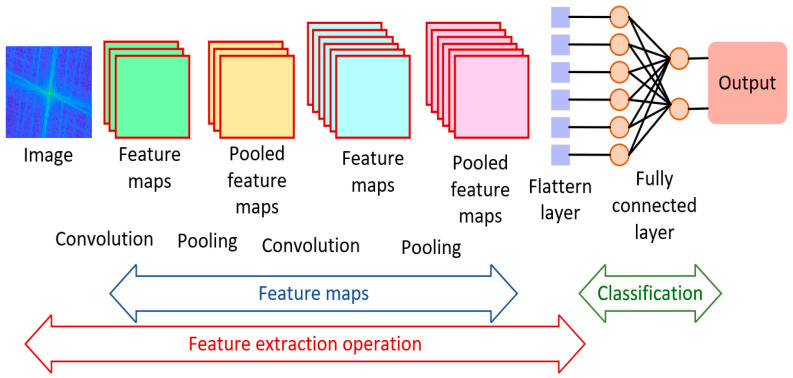
A typical convolutional neural network.

**Figure 2 healthcare-12-00994-f002:**
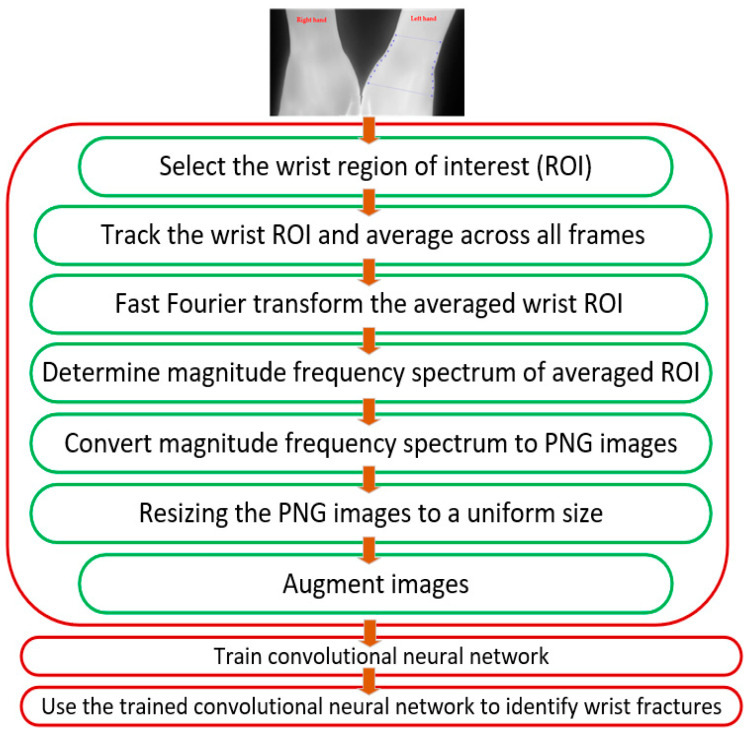
Operations to identify wrist fracture.

**Figure 3 healthcare-12-00994-f003:**
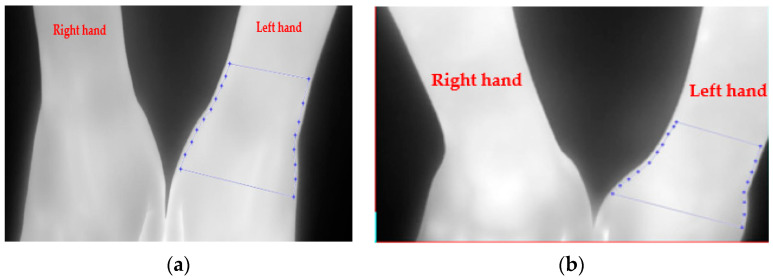
Infrared image of a participants’ hands: (**a**) fracture of left wrist; (**b**) sprained left wrist. The region of interest is shown by the blue dotted line.

**Figure 4 healthcare-12-00994-f004:**
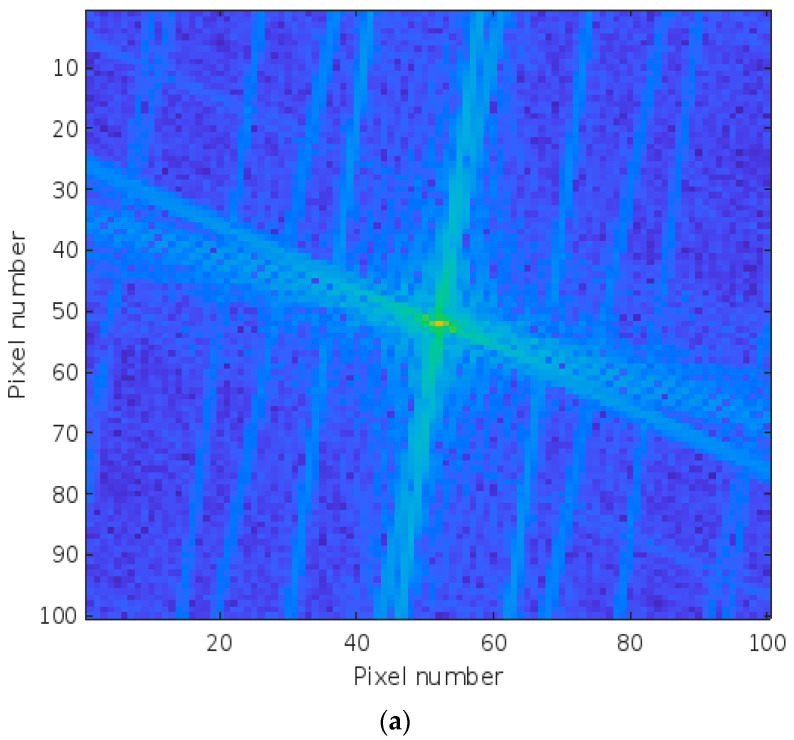
Magnitude frequency spectra of a typical region of interest for (**a**) fractured wrist ROI and (**b**) sprained wrist ROI.

**Figure 5 healthcare-12-00994-f005:**
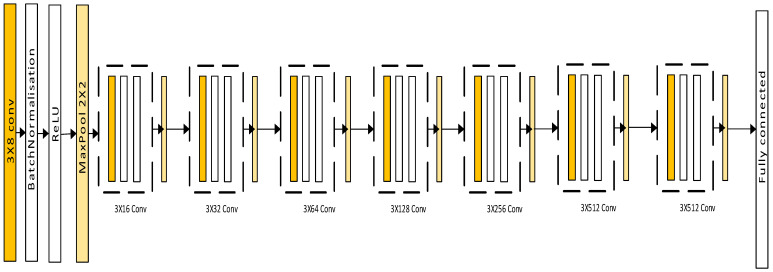
CNN-based deep neural network used in the study.

**Figure 6 healthcare-12-00994-f006:**
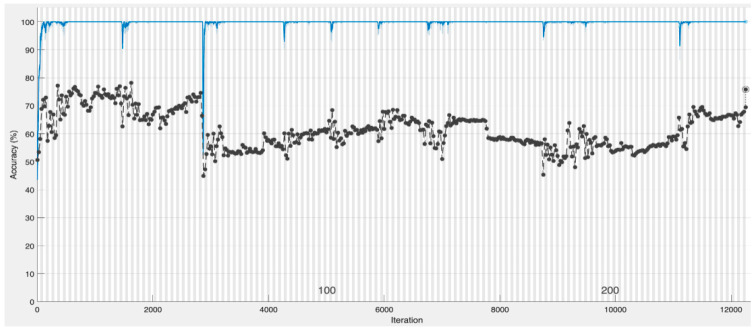
The CNN model training with validation data. The blue graph represents the model’s training, and the black graph represents the validation using the validation dataset with validation carried out after 250 epochs.

**Figure 7 healthcare-12-00994-f007:**
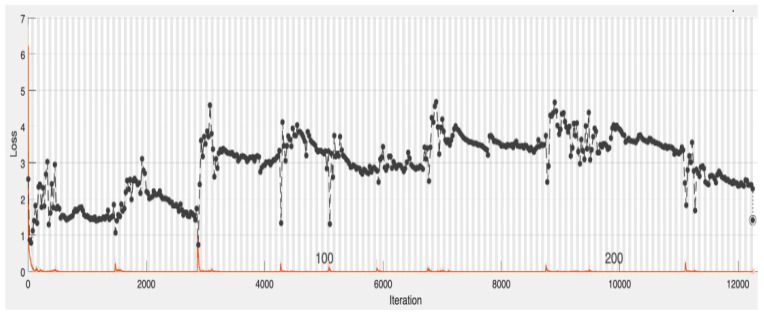
The CNN model loss with validation data. The red graph represents the model training loss (or error) and the black graph represents the validation error using the validation dataset with validation carried out after 250 epochs.

**Figure 8 healthcare-12-00994-f008:**
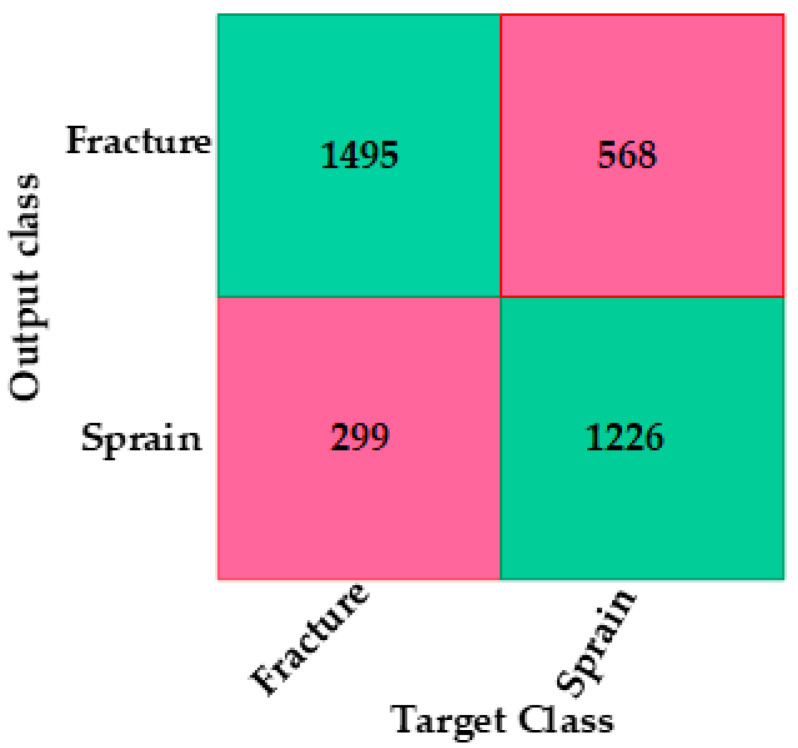
Confusion matrix for CNN-based deep neural network for wrist fracture versus sprain. The squares contain the number of images for validation dataset, total: 3588 images.

**Figure 9 healthcare-12-00994-f009:**
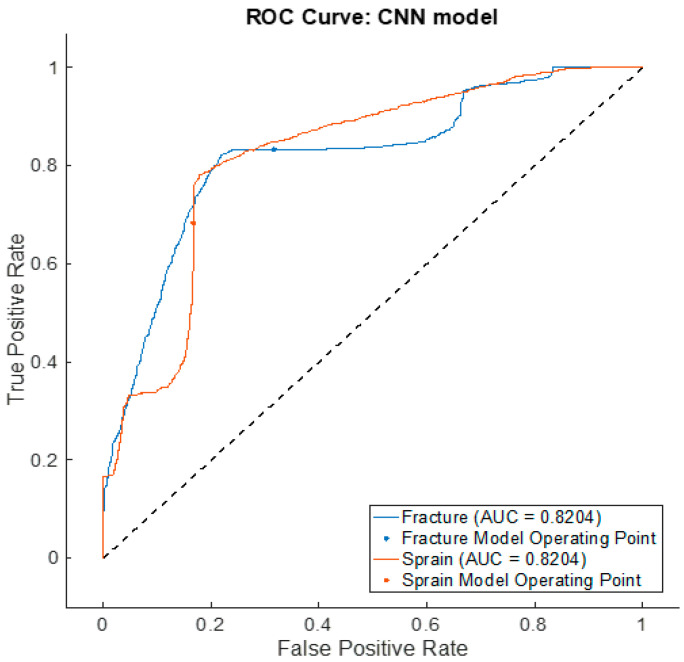
Receiver operating characteristic curve for CNN-based deep neural network.

**Table 1 healthcare-12-00994-t001:** Summary of related studies utilizing IRT imaging for medical diagnosis.

Study	Diagnosis	Number of Subjects	Region of Interest	Detected Temperature Difference (°C)
[32]	Bone neoplasia	40	Limb	0.53 to 0.67
[33]	Fracture	19	Forearm	0.8 to 2
[34]	Repetitive strain injuries	33	Wrist and hand Joints	0.65
[35]	Fracture/sprain	133	Various	0.26 to 0.47
[31]	Toddler’s fracture	39	Tibia	1.1

**Table 2 healthcare-12-00994-t002:** Summary of infrared thermal imaging-based machine learning methods for medical diagnosis.

Study	Number of Subjects	Diagnosis	ML/AI Algorithm	Accuracy (%)
[49]	283	Diabetic eye disease	SVM	86
[50]	11	Lip lesions	SVM/KNN	63–100
[51]	67	Breast cancer	Various	73–92
[52]	82	Pressure injury	CNN	95

**Table 3 healthcare-12-00994-t003:** Layer names and their learnable parameters.

Layer Name	Parameters
2D convolutional (Conv_1 (2D))	80
Batch normalization (Batchnorm_1 (2D))	16
2D Convolutional (Conv_2 (2D))	1168
Batch Normalization (Batchnorm_2 (2D))	32
2D Convolutional (Conv_3 (2D))	4640
Batch Normalization (Batchnorm_3 (2D))	64
2D Convolutional (Conv_4 (2D))	18,496
Batch Normalization (Batchnorm_4 (2D))	128
2D Convolutional (Conv_5 (2D))	73,856
Batch normalization (Batchnorm_5 (2D))	256
2D Convolutional (Conv_6 (2D))	295,168
Batch Normalization (Batchnorm_6 (2D))	512
2D Convolutional (Conv_7 (2D))	1,180,160
Batch normalization (Batchnorm_7 (2D))	1024
2D Convolutional (Conv_8 (2D))	1,180,160
Batch normalizations (Batchnorm_8 (2D))	1024
Fully Connected (FC)	1026

**Table 4 healthcare-12-00994-t004:** Hyperparameters for CNN-based deep neural network.

Hyperparameters	Measures
Optimizer	Adam
Activation function	ReLU
Learning rate	0.005
Batch size	170
Maximum epoch	250 (no early stopping)
Maximum iteration	12,250
Loss function	Cross-Entropy

**Table 5 healthcare-12-00994-t005:** Number of images used for CNN training and validation.

Injury	Training	Validation
Fracture	3887	1794
Sprain	4485	1794

**Table 6 healthcare-12-00994-t006:** Evaluation metrics for CNN-based deep neural network with augmentation and without dropout.

Injury Type	Sensitivity (%)	Specificity (%)	NPV (%)	PPV (%)	Accuracy (%)
Fracture versus sprain	88.3	68.3	80.4	72.5	75.8

**Table 7 healthcare-12-00994-t007:** Evaluation metrics for CNN-based deep neural network without augmentation and without dropout.

Injury	Sensitivity (%)	Specificity (%)	NPV (%)	PPV (%)	Accuracy (%)
Fracture versus sprain	50	66.7	60.0	57.1	58.3

## Data Availability

Patient data are not shared due to ethical restrictions.

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
