# Peer review of "Convolutional Neural Network to Classify Infrared Thermal Images of Fractured Wrists in Pediatrics"

_healthcare, 2024, doi:10.3390/healthcare12100994_

Round 1
Reviewer 1 Report
Comments and Suggestions for Authors
Regarding the manuscript with the title
"Convolutional Neural Network to Classify Infrared Thermal Images of Fractured Wrists in Pediatrics"
where in he authors developed and evaluated a CNN model for the classification of infrared thermal images to detect wrist fractures in children. Infrared thermal images were recorded from 19 participants with fractures and 21 participants without fractures, which were confirmed by X-ray radiography. 299 frames were recorded for each participant, resulting in 11,960 frames. These frames were FFT and resized from the center of the image to 100x100 pixels. Image enhancement techniques were used to improve CNN generalization. The CNN model had 34 layers and had 88% sensitivity and 76% accuracy in identifying wrist fractures.
This study has covered a scientific chat. The content is well organized. From a scientific point of view, this study is scientific and has its own attractions for clinical experts and scientific engineers. Its methodology is attractive and its method is scientifically presented. But there are still some points to be observed so that by applying these points, we can claim that this manuscript is suitable for publication.
1: The abstract is incomplete and needs to be completed. Please mention all the findings in the abstract.
2: In the introduction section, please dedicate a limited and small section to artificial intelligence and its applications in medical sciences and image and medical data analysis. For this purpose, it is better to add more resources in this section. You can also use the following sources.
"Automated detection model in classification of B-lymphoblast cells from normal B-lymphoid precursors in blood smear microscopic images based on the majority voting technique"
"Comparison of Feature Extraction with PCA and LTP Methods and Investigating the Effect of Dimensionality Reduction in the Bat Algorithm for Face Recognition"
"Mobile applications in COVID-19 detection and diagnosis: an efficient tool to control the future pandemic; a multidimensional systematic review of the state of the art"
Using these references can bring more and better resources for researchers.
3: Figure 1 is not a good representative for CNN networks. You can find more suitable images in the above sources and use them.
4: In Figure 2, use better writing. For example, instead of "Augment image", use "Image Augmentation" and Image resizing etc.
It seems that by fixing these revisions, this manuscript can be published.
Author Response
27 April 2024
Dear Honorary Editors, Respected Reviewers
Thank you very much for so kindly reviewing our article and making very valuable constructive comments. We have very carefully considered your comments and have done our best to amend and improve the paper according to your suggestions. The changes made are summarised in the table (attached) and are highlighted blue on the article.
We are very grateful for the help and support provided and hope our revisions meet your expectations.
The revised paper has now significantly improved.
Best wishes
Professor Reza Saatchi

Reviewer 2 Report
Comments and Suggestions for Authors
I reviewed this manuscript twice. It seems that the methodology of this manuscript is interesting. Good content has been presented and it has a moderate novelty that can be suggested for publication with a series of revisions. So, in my opinion, we should give the authors of this study a chance to consider these points in the revised version.
1: The abstract is not rich and badly needs correction. There are more findings in this study that the authors did not mention in the abstract. So it's better to write the abstract richer and more fruitful.
2: Why didn't you talk about implementation? What tool did you use? Why didn't you share the implementation codes or share the link?
3: Figure two is not suitable at all. Better fix it. Change its legend too. This figure is a general suggested model for your study, not image processing processes.
4: The sources of this manuscript are not enough. I suggest you use more sources that process medical images using CNN algorithms. For example, the following two studies can be considered.
Toward artificial intelligence (AI) applications in the determination of "COVID-19 infection severity: considering AI as a disease control strategy in future pandemics"
"An intelligent modular real-time vision-based system for environment perception."
The rest is well written and seems to be of high quality.
I hope the authors will use this opportunity and apply all the mentioned revisions in this manuscript.
Author Response

(The authors gave the same response as above.)

Reviewer 3 Report
Comments and Suggestions for Authors
The paper presents a DL CNN driven method for classifying wrist fractures in children. The subject is of interest however the authors should address a number of points before resubmitting.
1) There is no novelty in this paper. The DL method is a basic method. Why did the authors only used a CNN approach and did not use a VAE approach which could also be used to augment data?
2) Why did the authors used 5 CNN layers? What filters were applied at each stage?
3) The results do not show high accuracies. There is a lack of discussing the reason the results are not that promising. There is little discussion on the limitations of the study, which should be expanded in the discussion section.
4) How can this approach be used in another set of analogous data? The smaple of 19 patients is small. As noted before data augmentation in this case is a must. The methodology does not address convincingly this issue.
Comments on the Quality of English Language
Language is OK.
Author Response

(The authors gave the same response as above.)

Round 2
Reviewer 3 Report
Comments and Suggestions for Authors
The authors have not convincingly addresses the reviewers' comments. They refer now in the text on the innovation aspects they could have ysed but I asked them to apply first these techniques, evaluate the results and then resubmit. The augmentation section is weak and really not addressed. If the paper is to have any impact at all these suggestions should be implemented in the methodology, evaluated and then rewrite the paper.
Comments on the Quality of English LanguageEngish is OK.
Author Response
02 May 2024
Dear Honorary Editors, Respected Reviewer
Thank you very much for so kindly reviewing our article and making very valuable constructive comments. We have very carefully considered your comments and have done our best to amend and improve the paper according to your suggestions. The changes made are summarised in the table below and are highlighted blue on the article.
We are very grateful for the help and support provided and hope our revisions meet your expectations.
The revised paper has now significantly improved.
Best wishes
Professor Reza Saatchi
